# Revisiting Treatment of Metastatic Urothelial Cancer: Where Do Cisplatin and Platinum Ineligibility Criteria Stand?

**DOI:** 10.3390/biomedicines12030519

**Published:** 2024-02-26

**Authors:** Mohammad Jad Moussa, Matthew T. Campbell, Omar Alhalabi

**Affiliations:** Department of Genitourinary Medical Oncology, Division of Cancer Medicine, The University of Texas MD Anderson Cancer Center, Houston, TX 77030, USA; mmoussa1@mdanderson.org (M.J.M.); mcampbell3@mdanderson.org (M.T.C.)

**Keywords:** metastatic urothelial cancer, cisplatin-based chemotherapy, platinum-based chemotherapy, cisplatin ineligible, platinum ineligible, enfortumab vedotin, antibody-drug conjugates, immunotherapy combinations, immune checkpoint inhibitors, immunotherapy, frontline therapy

## Abstract

Cisplatin-based chemotherapy has been the standard of care in metastatic urothelial cancer (mUC) for more than two decades. However, many patients with comorbidities cannot receive cisplatin or its alternative, carboplatin. ‘Cisplatin-ineligible’ and ‘platinum-ineligible’ patients lacked effective therapy options. However, the recent combination of enfortumab vedotin (EV), an antibody–drug conjugate targeting Nectin-4, with pembrolizumab (P), an antibody targeting the programmed death-1 (PD-1) immune checkpoint, is changing the status quo of frontline mUC treatment, with potential synergy seen in the EV-103 and EV-302 clinical trials. First, we review the working definitions of ‘cisplatin ineligibility’ and ‘platinum ineligibility’ in mUC clinical trials and the standard of care in both categories. Then, we review select clinical trials for frontline treatment of cisplatin- and platinum-ineligible mUC patients on ClinicalTrials.gov. We classify the investigated drugs in these trials by their therapeutic strategies. Alongside chemotherapy combinations, the field is witnessing more immunotherapy combinations with fibroblast growth factor receptor (FGFR) inhibitors, bicycle toxin conjugates, bispecific antibodies, innovative targeted therapies, and many others. Most importantly, we rethink the value of classifying patients by cisplatin or platinum ineligibility in the frontline setting in the post-EVP era. Lastly, we discuss new priority goals to tailor predictive, monitoring, and prognostic biomarkers to these emergent therapies.

## 1. Introduction

Metastatic urothelial cancer (mUC) is an aggressive malignancy with limited treatment options. The recent phase 3 EV-302 study of combination enfortumab vedotin (EV) plus pembrolizumab (P) demonstrated impressive survival outcomes and response rates after a “stagnant” approach with frontline (1L) multiagent platinum-based chemotherapy for the last two decades [1,2]. Compared to a median survival of 16 months with cisplatin- or carboplatin-based regimens, overall survival (OS) and progression-free survival (PFS) have nearly doubled with this additive and potentially synergistic combination (EVP) [2]. The Food and Drug Administration (FDA) approved EVP for cisplatin-ineligible patients with locally advanced (LA) or metastatic urothelial carcinoma in April 2023 based on similar positive results from the EV-103 multicohort study [3]. In December 2023, full FDA approval of EVP was granted for all patients with LA/mUC [4].

In the era of chemotherapy, it was believed that 30% to 50% of patients with mUC were cisplatin ineligible [5,6,7]. Genitourinary oncology experts also estimated that 10% to 15% of patients with mUC were platinum ineligible, meaning they could not receive cisplatin or carboplatin [8]. Moreover, comorbidities often limited cisplatin eligibility, and survival with carboplatin-based regimens seemed inferior and limited (8–9 months) [9]. To optimize therapy for these populations in trials and clinical practice, Galsky et al. (2011) and Gupta et al. (2019) established much-needed standard definitions of “cisplatin ineligibility” and “platinum ineligibility”, respectively [7,10,11]. Yet, after the newly demonstrated benefit of EVP in cisplatin- and platinum-ineligible cohorts in phase 3 of EV-302, it is necessary to rethink the practical significance of these criteria in the 1L setting.

Immunotherapy combinations with antibody–drug conjugates (ADC), such as EV, which targets the preponderant expression of Nectin-4 in urothelial cancer, and sacituzumab govitecan, which targets Trop-2, are not the sole strategies in the booming new field of mUC therapies. Fibroblast growth factor receptor (FGFR) inhibitors, such as erdafitinib, have also been studied in cohorts with FGFR genomic alterations [12]. Other more recent drug classes, such as bispecific antibodies, bicycle toxin conjugates, and new immune-targeted and cellular therapies, are also being investigated in several mUC treatment settings [13]. In the era of precision medicine, understanding the relevance of biomarkers is crucial for predicting treatment response, toxicity, and optimal treatment duration. 

In this review, we compare the recent consensus criteria defining cisplatin and platinum ineligibility in mUC and elucidate the previous and current standard of care (SOC) in cisplatin- and platinum-ineligible patients. After reviewing the most relevant 1L clinical trials and treatment strategies in both categories, we discuss the future of classifying patients by chemotherapy fitness in the post-EVP era.

## 2. Materials and Methods

We conducted a comprehensive search on ClinicalTrials.gov, as last updated on 3 November 2023, using the query terms ‘metastatic urothelial carcinoma’ in the ‘Condition or disease’ field and ‘metastatic urothelial cancer’ in the ‘Other terms’ field. A flow diagram showing the yielded original database (*n* = 300) is shown in Figure 1. 

We excluded *n* = 77 studies that were suspended, terminated, withdrawn or of unknown status, and primarily included *n* = 223 studies that were completed, active but recruiting, active but not recruiting, or not yet recruiting. As our research interest in this narrative review is limited to studies in the 1L treatment setting of cisplatin-ineligible and platinum-ineligible LA/mUC, we reviewed the study details of the *n* = 223 studies and excluded all trials (*n* = 177) not including this treatment setting in these two populations in any of their cohorts.

At the last step of data refinement, we conducted a thorough individual review of the inclusion and exclusion criteria of the included 46 trials, to separate them into those including ‘cisplatin-ineligible’ LA/mUC patients only, ‘platinum-ineligible’ LA/mUC patients only, or both populations. Finally, we cross-referenced the included trials with published papers or abstracts with reportable outcomes (primary, partial, or complete results) to summarize outcomes of interest and discuss their clinical implications for the genitourinary oncologist. 

## 3. Definitions of Cisplatin Ineligibility and Platinum Ineligibility

### 3.1. Definition of Cisplatin Ineligibility

Patients with treatment-naïve mUC can be classified into three categories: cisplatin eligible, cisplatin ineligible but carboplatin eligible, and platinum ineligible (cisplatin and carboplatin ineligible). Up to half of patients with mUC meet one common criterion for cisplatin ineligibility [6]. In one of the widest efforts to define cisplatin ineligibility in mUC, Galsky et al. (2011) proposed a working group definition of cisplatin-ineligible mUC, the criteria for which can be found in Table 1 [7]. The presence of one criterion out of five is enough to establish ineligibility.

A possible exception may occur if a patient’s sole criterion for ineligibility is borderline renal function (creatinine clearance of 40–60 mL/min) [13]. In that case, one notable strategy is a dose split of cisplatin with short duration and enhanced hydration, which has demonstrated increased nephroprotective effects and possibly similar efficacy in prospective studies with no comparator arms [14,15].

### 3.2. Definition of Platinum Ineligibility

Platinum ineligibility automatically entails ineligibility for both cisplatin and carboplatin. In clinical practice, carboplatin ineligibility is often determined using physicians’ common sense, mainly based on an assessment of overall performance status and renal function. Many oncologists would prefer not to prescribe carboplatin to frail elderly patients with limited physiological reserves [13] and frequent comorbidities, such as worsening heart disease and uncontrolled diabetes mellitus (DM).

The lack of a formal definition of carboplatin ineligibility led to the first initiative by Gupta et al. (2019) to survey genitourinary oncologists about their experiences and choices [10]. The need for a formal definition became more urgent when the FDA restricted the use of pembrolizumab and atezolizumab to cisplatin-ineligible patients with tumors with high expression of programmed death ligand 1 (PD-L1) or platinum-ineligible patients regardless of PD-L1 status, despite the fact that both treatments had previously been approved unconditionally for 1L treatment of cisplatin-ineligible patients [16]. 

In November 2022, the FDA approval of atezolizumab for 1L mUC was completely withdrawn, and pembrolizumab was restricted to 1L treatment of platinum-ineligible patients [17]. Thus, Gupta et al. (2022) surveyed 60 genitourinary medical oncologists in the United States and created an updated consensus definition of platinum ineligibility for mUC patients meeting at least one of five criteria, as shown in Table 1 [11]. The same group estimated a prevalence of <10% carboplatin ineligibility among LA/mUC patients [11].

## 4. Treatment of Cisplatin-Ineligible Metastatic Urothelial Carcinoma

### 4.1. Previous Standard of Care 

The ancillary role of cisplatin-based chemotherapy in LA/mUC was established more than three decades ago with MVAC (methotrexate, vinblastine, doxorubicin, and cisplatin), yielding a median OS (mOS) of 13 months compared to single-agent cisplatin [18]. Later, dose-dense (or accelerated) MVAC became an accepted option in clinical practice given its more favorable toxicity profile in the phase 3 EORTC 30924 trial, despite sharing a similar OS with standard MVAC [19]. However, gemcitabine plus cisplatin (GC), another frequent SOC treatment for LA/mUC, also showed response rates and survival outcomes similar to those of MVAC, with the exception of better tolerability and less toxicity in favor of GC [1]. 

In cisplatin-ineligible patients, gemcitabine plus carboplatin (GCa) was the 1L alternative to GC, based on the results of the historical trial EORTC 30986 [9]. Although GCa yielded a response rate of ~40%, the OS (usually ~9 months) was shorter than that of GC (15–16 months). After achieving stable disease with 4–6 cycles of platinum-based regimens, the landmark phase 3 JAVELIN Bladder-100 trial, in which 40% of enrolled patients received GCa, found significant OS benefit with avelumab as maintenance therapy regardless of PD-L1 status [20]. JAVELIN Bladder-100 was the first trial in more than 3 decades to prove the amelioration of survival outcomes for patients without disease progression, including stable disease, on 1L platinum-based chemotherapy. An OS benefit was attributed to avelumab compared to best supportive care (21.4 mo vs. 14.3 mo; Hazard Ratio (HR) = 0.69, 95% CI 0.56–0.86, *p* = 0.001). Despite the expected lower response rate with GCa, subgroup analyses of JAVELIN Bladder-100 consolidated the survival benefit of avelumab regardless of receipt of GC or GCa [21]. However, the study did not report the rate of primary progression on GCa prior to randomization, which would be anticipated to be higher than that of GC.

Other less frequently used regimens in the chemotherapy era replaced cisplatin with taxanes, such as paclitaxel or docetaxel, or even adopted single-agent chemotherapy (gemcitabine) [22,23]. However, no phase 3 trial involving these non-platinum-based regimens or sequential treatment doublets was performed in our population of interest.

### 4.2. The New Standard of Care 

Enfortumab vedotin (EV) is a breakthrough humanized monoclonal ADC targeting Nectin-4, a highly expressed protein in urothelial cancer [24]. It induces an anti-proliferative and pro-apoptotic effect on cancer cells through the release of monomethyl auristatin E (MMAE), a tubulin-toxic chemotherapeutic agent [24]. After being internalized into the cell to release MMAE [25], EV exhibits targeted cytotoxicity while minimizing systemic toxicity. 

Cohort K of the phase 2 study EV-103/KEYNOTE-869 randomized treatment-naïve and cisplatin-ineligible patients to receive EV, either alone or in combination with pembrolizumab (EVP). In the latest updates from this cohort, the combination arm achieved an objective response rate (ORR) of 64.5% and a complete response rate (CRR) of 10.5%, with a median duration of response (mDOR) not yet reached, compared to an ORR of 45.2% and mDOR of 13.2 months in the EV monotherapy group [26,27]. 

Despite no formal statistical comparison between the survival outcomes of the EV vs. EVP arms, the high, durable, and early-onset responses to EVP were unprecedented in the chemotherapy era. Interestingly, the overwhelming majority of patients enrolled in Cohort K had visceral disease, a negative prognostic factor and a Bajorin risk factor [28]. The percantage of patients with ECOG PS 2 was also balanced between treatment arms in this cohort with heavy metastatic burden. In subsequent analysis of Cohort K, EVP activity was consistently seen in subgroups with worse prognosis, especially patients with visceral metastases (ORR in EVP arm: 65.6% [52.7–77.1]) [27]. 

Additionally, a 4-year follow up of EV-103 dose escalation (Cohort A) consolidated the deep (ORR 73.1%, CRR 15.6%) and durable (mDOR: 22.1 months; mOS: 26.1 months) responses to EVP [29]. The safety profile in this follow-up was consistent with previous reports. While the 2019 and 2021 FDA approvals of EV concerned platinum- or immunotherapy-exposed patients [30], the latest accelerated approval in April 2023 covered treatment-naïve, cisplatin-ineligible patients [3].

More recently, phase 3 EV-302 confirmed the survival endpoints achieved with EVP vs. platinum-based chemotherapy (GC or GCa) [2]. EVP almost doubled the mOS (31.5 mo vs. 16.1 mo; HR 0.47; 95% CI: 0.38–0.58, *p* < 0.00001) and median PFS (mPFS) (12.5 mo vs. 6.3 mo; HR 0.45, 95% CI: 0.38–0.54, *p* < 0.00001) at a median follow-up of 17.2 months. The response rate achieved by EVP was also significantly higher (67.7% vs. 44.4%, *p* < 0.00001). Together, these findings have propelled EVP toward a “dethroning” of the stagnant SOC of chemotherapy.

The preference for using EVP over chemotherapy will likely be dictated by the interaction of the regimen’s toxicity profile with the patient’s medical comorbidities. Currently, there are no contraindications to EV in its official prescribing information. However, warnings and precautions have been issued for patients with preexisting DM and previous peripheral neuropathy [31]. Beyond these previously reported treatment-related adverse events (TRAEs), no additional safety signals for EV or pembrolizumab were reported in these trials.

## 5. Investigational Regimens in Cisplatin-Ineligible Metastatic Urothelial Carcinoma

### 5.1. Antibody-Drug Conjugates with or without Immune Checkpoint Inhibitors (ICIs)

Sacituzumab govitecan (SG) is another promising humanized monoclonal ADC targeting Trop-2, a highly expressed protein in urothelial carcinoma with roles in tumor cell proliferation, migration, and invasion [32]. Similar to EV, it is coupled with a chemotherapeutic agent that acts as a topoisomerase I inhibitor derived from irinotecan (SN38) via a hydrolysable linker [32]. After the phase 2 TROPHY-U-01 trial (NCT03547973), SG was FDA approved for patients with mUC previously treated with platinum-based chemotherapy or immunotherapy [33]. Cohort 6 of TROPHY-U-01 was designed for cisplatin-ineligible, treatment-naïve LA/mUC, with SC studied alone and in combination with either zimberelimab (anti-PD-1 monoclonal antibody) or zimberelimab + domvanalimab (anti-T-cell immunoglobulin and ITIM domain [anti-TIGIT] monoclonal antibody). 

Another phase 2 study (NCT04863885) is evaluating the combination of SG with ipilimumab (anticytotoxic T lymphocyte–associated protein 4 [anti-CTLA4]) and nivolumab (anti-PD-L1), coupled with biomarker analysis. Early indicators of activity in phase 1 were noted in a small cohort of 6 response-evaluable patients (1 complete response [CR] and 3 partial responses [PR]) [34]. In this study, the mDOR was 9.2 mo (range: 4.6–12 mo) and mPFS was 8.8 mo (95% CI: 3.8–Undefined).

### 5.2. FGFR-Targeted Therapies with or without Immune Checkpoint Inhibitors

FGFR3 protein, encoded *FGFR3* on chromosome 4, is a tyrosine kinase with multiple roles in development, osteogenesis, and bone maintenance [12]. Among all cancers, FGFR3 aberration occurs most frequently in urothelial cancer (18% of cases), particularly platinum-treated urothelial cancer [12,35]. Table 2 summarizes the most important clinical trials testing the combination of FGFR inhibitors and ICIs, inclusively or exclusively, in the 1L LA/mUC population.

Erdafitinib is currently the only FDA-approved FGFR1–4 inhibitor for mUC with FGFR2 or FGFR3 alterations after progression during or following platinum-based regimens [12,36]. Its approval marked the first use of gene-targeted therapy in bladder cancer. ICIs were previously assumed to lack efficacy in FGFR-altered urothelial cancers, such as luminal 1 tumors, which have more frequent FGFR3 alterations and a cold immune microenvironment [37]. However, this assumption contrasts with the recent clinical results of Cohort 2 of the randomized phase 3 THOR trial. Treatment with erdafitinib was not associated with improvement in mOS as compared to pembrolizumab in patients with FGFR-altered mUC that progressed after one prior therapy [38].

The phase 2 NORSE trial (NCT03473743) had the largest sample to investigate erdafitinib alone or with cetrelimab (anti-PD-1 agent) in the 1L setting. Recently, promising durable activity was reported with erdafitinib and cetrelimab (ORR 54.5%, CR 13.6%) compared to single-agent erdafitinib (ORR 44.2%, CR 2.3%) in cisplatin-ineligible patients with specific FGFR3 mutations and/or fusions [39]. Patients with ECOG PS 2 comprise about a third of response-evaluable patients (*n* = 87). Even in patients with low combined positive score (CPS < 10), ORR was reported to be 46.4% and 50% with erdafitinib alone and the combination, respectively [39]. However, given the time required to accrue a biomarker-required study, the negative findings from Cohort 2 of the THOR study, and the positive findings from the EV302 study, it is unlikely that a phase 3 study testing this combination will move forward at this time.

Other trials with other FGFR inhibitors have been less successful in the 1L setting. Phase 1b of FORT-2 (NCT03473756) studied rogaratinib, an oral pan-FGFR inhibitor (FGFR1–4), with atezolizumab in the 1L setting [40]. Among 26 patients with FGFR1/3 mRNA overexpression, 54% had an objective response, including 3 CR (13%) and 10 PR (42%) [41]. However, in the randomized phase 2/3 FORT-1 trial, rogaratinib alone failed to improve response rates and OS compared with dealer’s choice of chemotherapy in the subsequent-line setting (previously treated patients with similar FGFR alterations) [42]. The phase 1b/2 FIDES-2 trial, which aimed to evaluate derazantinib monotherapy at different doses, was halted in advance as a result of a suboptimal PFS and ORR of 8% when given as second- or third-line treatment in a cohort of previously treated mUC patients with FGFR1–3 genomic alterations [43]. The phase 2 FIGHT-205 trial evaluating pemigatinib, a selective FGFR1–3 inhibitor, alone and in combination with pembrolizumab was stopped due to a business decision [44]. 

### 5.3. Single-Agent or Combined Immune Checkpoint Inhibitors

Single-agent ICIs have proven less effective in LA/mUC, with lower response rates and higher rates of primary progression. Avelumab was approved as maintenance therapy after 1L platinum-based treatment in LA and mUC [20]. It was also studied alone in a single-arm phase 2 trial (ARIES) in the 1L setting in cisplatin-ineligible patients with positive PD-L1 expression. After a median follow-up of 9 months, the ORR was suboptimal (22.5%), with no difference in 1-year OS between patients with CPS < 10 and those with CPS ≥ 10 [45].

DANUBE was a phase 3 trial (NCT02516241) comparing single-agent durvalumab (anti-PD-L1 agent), combination durvalumab and tremelimumab (anti-CTLA-4), and chemotherapy. The main goals were to compare OS between durvalumab and GC/GCa in patients with tumors with high PD-L1 expression (≥25%) and then between combination durvalumab and tremelimumab and GC/GCa regardless of PD-L1 expression. While neither primary endpoint was met, a secondary endpoint of OS favored the combination versus GC/GCa in patients with high PD-L1 expression (HR 0.74 [95% CI: 0.59–0.93]) [46]. These findings prompted an update to the design of the phase 3 NILE trial, discussed in the next section, to focus on this subpopulation with high PD-L1 expression. 

The combination of nivolumab and ipilimumab, studied in the phase 3 CheckMate-901 trial, also failed to meet the OS endpoint in treatment-naïve, surgery-ineligible mUC patients with tumors expressing PD-L1 ≥ 1% [47]. However, the results are yet to be announced for the cisplatin-ineligible subgroup. To note, another experimental arm in CheckMate-901, consisting of 6 cycles of GC with maintenance nivolumab, was compared with the control arm (GC) in cisplatin-eligible patients. The study met OS and PFS endpoints for this combination, with an ORR of 57.6% covering a CR rate of 22% [48].

### 5.4. Chemotherapy with or without Other Drug Classes

#### 5.4.1. Chemotherapy + Immunotherapy

Phase 2 KEYNOTE-052 reported the efficacy of single-agent pembrolizumab in treatment-naïve, cisplatin-ineligible patients from 20 countries [49]. In the 2020 analysis, the survival benefit was further highlighted with high PD-L1 expression, defined by CPS ≥ 10 [50]. Yet, phase 3 KEYNOTE-361 did not demonstrate an OS benefit with pembrolizumab and chemotherapy compared to chemotherapy alone [51]. 

Atezolizumab was initially granted accelerated approval by the FDA, based on some clinical activity reported in phase 2 IMvigor210 [52]. However, in randomized phase 3 IMvigor130, atezolizumab combined with GCa did not confer a longer OS, despite an improvement in PFS [53]. Therefore, the FDA withdrew the regulatory approval of both pembrolizumab and atezolizumab for patients with cisplatin-ineligible mUC [16]. 

The ongoing phase 3 NILE study (NCT03682068) is designed to randomize more than 1000 patients into 3 arms (1:1:1): durvalumab and chemotherapy (including GCa if cisplatin ineligible) in Arm 1; durvalumab, tremelimumab, and GCa in Arm 2; and GCa alone in Arm 3 [54]. Following the results of the DANUBE trial, the primary endpoint will be updated to compare OS in patients with high PD-L1 expression in Arm 1 vs. Arm 3 and in Arm 2 vs. Arm 3 [54]. 

Since avelumab showed OS improvement as maintenance therapy for patients not progressing on 1L chemotherapy [20,21], 1L avelumab with GCa was tested in the phase 2 INDUCOMAIN trial (NCT03390595). The first study arm received induction avelumab, then combination avelumab and GCa, then avelumab alone until progression or intolerance. The second arm received GCa only. No statistical difference was noted in PFS or OS between the two arms, with a high rate of early progression (31%) with induction avelumab versus the control group (9.3%) [55].

#### 5.4.2. Chemotherapy + Kinase Inhibitors

Trilaciclib, a CDK4/6 inhibitor, is FDA approved to reduce chemotherapy-induced myelosuppression in patients with extensive-stage small cell lung cancer [56]. PRESERVE-3 (NCT04887831) is an exploratory phase 2 trial assessing whether olaparib added to chemotherapy (GC or GCa) enhances anti-tumor efficacy and lowers chemotherapy-induced myelosuppression. The study design includes no formal definition of cisplatin ineligibility, but the trial concerns a cohort of treatment-naïve patients with mUC and ECOG PS ≤ 2. The cohort will be randomized 1:1 to receive GC or GCa with or without olaparib. Maintenance avelumab, with and without olaparib, will be offered to patients until disease progression, trial end, major side effects, or investigator/patient decision. 

Currently, bevacizumab, the only anti-angiogenic agent (anti-VEGF) that has been studied in cisplatin-ineligible mUC, has no indication in any line of treatment. A phase 2 trial (NCT00588666) that studied GCa with bevacizumab showed a response rate of 49% [57]. Yet, in subsequent phase 3 trials, the combination only improved PFS, not OS [58].

#### 5.4.3. Other Non-Platinum Chemotherapy

Vinflunine has regulatory approval for cisplatin-ineligible patients who progressed on immunotherapy in Europe, but not in the United States [59]. In the phase 2/3 VINGEM study (NCT02665039), combination vinflunine plus gemcitabine failed to improve PFS or OS compared to GCa [60]. 

The phase 1 AVETAX trial (NCT03575013) also studied docetaxel plus avelumab in treatment-naïve cisplatin-ineligible patients with mUC, as well as those progressing during/after GC or GCa or within 12 months of platinum-based neoadjuvant/adjuvant chemotherapy [61]. At a proven safe dose of 75 mg/m^2^, ORR was 70% (CR 30%, PR 40%), mPFS was 9.2 months, and mOS was not reached in a cohort of 20 response-evaluable patients.

### 5.5. Tyrosine Kinase Inhibitors (TKIs) with or without Immune Checkpoint Inhibitors

Cabozantinib, a multi-kinase inhibitor of MET, AXL, and VEGFR2, can enhance tumor response to immune checkpoint inhibitors [62]. In this review, we limit our interest to the combination of cabozantinib plus atezolizumab in Cohort 3 of the phase 1b COSMIC-021 trial. With a mDOR of 7.1 months, the ORR was 20% and the disease control rate (DCR) was 80% [63]. Median PFS was 5.6 months and mOS was 14.3 months. 

Phase 3 LEAP-011 (NCT03898180) compared 1L lenvatinib plus pembrolizumab to pembrolizumab monotherapy in cisplatin-ineligible patients with CPS ≥ 10 [64]. No difference in ORR or survival outcomes (mOS and mPFS) was observed between the arms.

### 5.6. Bicycle Toxin Conjugates (BTCs) with Immune Checkpoint Inhibitors

Bicyclic peptides, a new class of polypeptides with antibody-like affinity and selectivity, are emerging as revolutionary target-binding drugs [65]. A phase 1/2 trial (NCT04561362) is studying BT8009, a BTC that selectively binds Nectin-4, as a 1L option for cisplatin-ineligible patients in Cohort B-7 [66]. BT8009 is a hydrophilic peptide drug that swiftly moves from the bloodstream through tissues to access and target cancer cells [67]. These distinctive characteristics may set it apart from ADCs in terms of tumor penetration and minimization of systemic exposure and toxicity. In phase 1, at a weekly recommended phase 2 dose (RP2D) of 5 mg/m^2^, ORR was 50% and DCR was 75% (1 CR, 3 PR, and 2 SD [stable disease]) in 8 response-evaluable patients, and at a weekly dose of 2.5 mg/m^2^, ORR was 25% and DCR was 75%. An expansion phase will evaluate BT-8009 as monotherapy and in combination with pembrolizumab at 2 RP2Ds in in our population of interest [68].

### 5.7. Other Drugs with Immune Checkpoint Inhibitors

Other drugs, mainly with a backbone of immunotherapy, have also been investigated in the LA/mUC 1L setting. A phase 2 trial (NCT04486781) studied sEphB4, a recombinant fusion protein of soluble Ephrin-B4 and albumin, with pembrolizumab. Among pretreated patients who experienced disease recurrence or progression on GC or GCa and whose tumors expressed Ephrin-B2, ORR and CRR were 52% and 24% respectively, with a mOS of 22 months [69]. The final analysis of response rates in phase 2 PIVOT-10 (NCT03785925) showed that the combination of bempegaldesleukin, a PEGylated interleukin-2 [IL-2] agent, with nivolumab did not reach contemporary efficacy benchmarks. Other investigated strategies include IO102-IO103 (immune-modulatory vaccines) with pembrolizumab (NCT05077709), tocilizumab (interleukin-6 [IL-6] receptor antagonist) with ipilimumab and nivolumab (NCT04940299), and sonidegib (hedgehog pathway inhibitor) with pembrolizumab (NCT04007744).

### 5.8. Radiation Therapy and Immune Checkpoint Inhibitors

In patients with LA/mUC, radiation therapy is applied as palliative treatment at the discretion of the treating physician [70]. It might offer limited symptom control of brain and bone metastasis or symptomatic primary bladder tumors. Nevertheless, interest in the synergistic action of immunotherapy and radiation therapy comes from the abscopal effect in some preclinical and early clinical trials [71]. Currently, a phase 2 trial (NCT03601455) is evaluating radiation therapy and durvalumab, with or without tremelimumab in patients with LA/mUC. In the safety lead-in cohort, which comprised pretreated patients who received maximal TURBT followed by durvalumab 1500 mg IV every 4 weeks and bladder stereotactic body radiotherapy (SBRT) of 33 Gy in 5 fractions between the first two cycles, the DCR was 70% and the local control rate was 90% [72].

## 6. Treatment of Platinum-Ineligible Metastatic Urothelial Carcinoma

### 6.1. Previous Standard of Care 

With the stagnant absence of any agent shown to have better efficacy than cisplatin- and carboplatin-based regimens, platinum-ineligible LA/mUC patients had an unmet therapeutic need for more than two decades. The most compelling indication for pembrolizumab in this population comes from phase 2 KEYNOTE-052 [50]. Even after a median follow-up of almost 5 years, 1L pembrolizumab conferred lasting clinical response, with an ORR of 28.9%, which was even higher for patients with CPS ≥ 10%. Median OS was 11.3 months and the 12- and 24-month OS rates were 46.9% and 31.2%, respectively [50]. Based on these results, the FDA granted approval for pembrolizumab as 1L treatment for treatment-naïve, platinum-ineligible patients with mUC [16]. 

Atezolizumab was previously granted accelerated FDA approval, based on results from phase 2 Imvigor210 in cisplatin-ineligible patients [52]. Later, the results of Phase 3 Imvigor130 and KEYNOTE-361, comparing chemotherapy to ICI monotherapy, showed that atezolizumab lacks clinical benefit in cisplatin-eligible patients with low or negative PD-L1 expression [53]. Thus, atezolizumab was limited to cisplatin-ineligible patients with PD-L1+ tumors (≥5% expression in immune cells) and platinum-ineligible patients regardless of PD-L1 expression [15,16]. The combination of atezolizumab and either GC or GCa also failed to meet a co-primary endpoint of OS benefit in phase 3 Imvigor 130 and only showed a PFS benefit [53]. After withdrawal of the indication of atezolizumab by the manufacturing company, approval was withdrawn in the US for platinum-ineligible patients regardless of PD-L1 status [17].

### 6.2. The New Standard of Care 

As discussed earlier, the superior outcomes of EVP compared with platinum-based chemotherapy, regardless of fitness to receive platinum compounds, support the use of this combination for 1L treatment of platinum-ineligible patients [2].

## 7. Investigational Regimens in Platinum-Ineligible Metastatic Urothelial Carcinoma

An overview of the ongoing registered clinical trials in platinum-ineligible mUC, highlighting the main investigated therapeutic strategies, is shown in Table 3.

### 7.1. FGFR Inhibitors with Immune Checkpoint Inhibitors

A phase 2 trial (NCT04601857) is studying the FGFR1–4 inhibitor futibatinib in combination with pembrolizumab in the 1L setting in our population of interest. The eligible population is split into two cohorts, A and B, which represent, respectively, patients with an FGFR3 mutation or FGFR1–4 fusion/rearrangement and those with other FGFR or non-FGFR genetic aberrations or wild-type tumors. Interest in this combination is based on early safety lead-in results supporting its safety and tolerability [73]. 

### 7.2. Targeted Therapies Alone or with Immune Checkpoint Inhibitors

LEAP-011 (NCT03898180) is a phase 3 trial testing the combination of the TKI lenvatinib with pembrolizumab in cisplatin-ineligible patients with PD-L1+ tumors and in platinum-ineligible patients regardless of PD-L1 expression [74]. There was no statistical difference in survival indicators (PFS and OS) between pembrolizumab plus lenvatinib and pembrolizumab plus placebo.

The randomized phase 2 BAYOU trial (NCT03459846) aimed to evaluate the poly (ADP-ribose) polymerase (PARP) inhibitor olaparib with the anti-PD-L1 agent durvalumab or placebo [75]. Although the primary endpoint, PFS, was not significantly different between both arms overall, a significant PFS difference was found in sub-analysis after stratification by HRR mutation status, in favor of the group harboring the mutation.
biomedicines-12-00519-t003_Table 3Table 3Overview of clinical trials registered on ClinicalTrials.gov by November 2023 that include patients with treatment-naïve platinum-ineligible mUC in their enrollment. Abbreviations: EZH2: enhancer of zeste homolog 2; FGFR: fibroblast growth factor receptor; N/A: not assessed; NCT: national clinical trial; ORR: objective response rate; OS: overall survival; PD-L1: programmed death ligand 1; PFS: progression-free survival; RP2D: recommended phase 2 dose; mUC: metastatic urothelial cancer.NCT Number + Title (If Available)Phase + EnrollmentStudy StatusCohorts/Arms of Interest for Treatment-Naïve Platinum-Ineligible mUCBiomarkerPrimary EndpointsNCT02573259Phase 1(147)Completed5 arms with same treatment (PF-06801591) but increasing concentrations and different doses of administrationPD-L1Parameters related to adverse events in Part 1, ORR in Part 2 [76]NCT04601857Phase 2(46)Active, recruitingCohort A: Futibatinib and Pembrolizumab, for patients with a FGFR3 mutation or FGFR1-4 fusion/rearrangement.Cohort A: *FGFR3* mutation or *FGFR1-4* fusion/rearrangement. ORR [77]Cohort B: Same treatment, but for all other patients than in Cohort A with UC (including patients with other FGFR or non-FGFR genetic aberrations and patients with wild type [non-mutated] tumors).Cohort B: other *FGFR* or non-*FGFR* genetic aberrationsNCT04486781Phase 2(38)Active, recruitingCombination therapy for all Ephrin B2ORR [78]NCT05645692Phase 2(240)Active, recruitingArm A (Atezolizumab) Q3W, Arm B (IV RO7247669) Q3W and Arm C (IV RO7247669 and tiragolumab) Q3WPD-L1ORR [79]NCT03854474Phase 1|Phase 2(30)Active, recruitingExperimental: Treatment (tazemetostat, pembrolizumab)*EZH2* and H3K27me3 chromatin methylationRP2D [80]NCT03898180[LEAP-011]Phase 3(487)Active, not recruitingExperimental: Pembrolizumab + Levantinib; Active Comparator: Pembrolizumab + Placebo; Experimental: Pembrolizumab monotherapyN/APFS and OS [81]NCT03288545[EV-103]Phase 1|Phase 2(348)Active, not recruitingCohort K: Enfortumab Vedotin + PembrolizumabN/AORR (Cohort K only) [82]


Enhancer of zeste homolog 1/2 (EZH1/2) inhibitors are also being considered for patients with platinum-ineligible mUC. The *EZH2* gene and its backup *EZH1* belong to a family of genes that are epigenetic regulators or repressors of transcription, primarily regulating cell cycle progression, autophagy, apoptosis, and senescence [83]. Thus, mutation and abnormal expression of these genes could be a driving force of metastasis. Examples include a currently active but not recruiting trial (NCT03854474) studying EZH2 inhibitor tazemetostat combined with pembrolizumab. A recruiting phase 1 trial (NCT04388852) studying the combination of tazemetostat with ipilimumab in a cohort of metastatic prostate, urothelial, and kidney cancers also includes platinum-ineligible patients in its enrollment.

### 7.3. Bispecific Antibodies with Immune Checkpoint Inhibitors

Bispecific antibodies are engineered to bind to two epitopes simultaneously, allowing them to modulate multiple signaling mechanisms [84]. The anti-PD1/anti-LAG3 bispecific antibody tobemstomig (RO7247669) [85] is currently being studied in a phase 2 trial (NCT05645692) in combination with atezolizumab. This antibody exerts a cytotoxic T lymphocyte–mediated response by inhibiting the PD-1– and LAG3-mediated downregulation of the activation and multiplication of T cells [86].

### 7.4. Recombinant Fusion Proteins with or without Immune Checkpoint Inhibitors

Human serum albumin (HSA)-based drugs are ‘tumor-oriented therapies’ with remarkable efficacy in drug delivery and biocompatibility, resulting in reduced toxicities [87]. One combination of HSA with a recombinant fusion protein, soluble EphB4-human serum albumin (sEphB4-HSA), is of potential interest in ongoing clinical trials in mUC. Interest in this combination is driven by the high expression of Ephrin-B4 in urothelial carcinoma and its driving role in tumor angiogenesis through the activation of its target protein, Ephrin-B2 [69]. Soluble EphB4 binds and sequesters Ephrin-B2 and arrests bidirectional signaling between PI3K-AKT and MAPK, thus exerting an anti-growth effect and attracting immune cells into the tumor [88]. This goal could be achieved by preventing T cell exhaustion, so an immunotherapy combination would be a booster strategy. Hence, a phase 2 trial (NCT04486781) is studying sEphB4 plus pembrolizumab, with a mOS of 14.6 months and mPFS of 4.1 months in platinum-exposed patients with recurrent or progressing mUC [69]. Interestingly, for patients with Ephrin B2–expressing tumors (~67% of enrolled patients), ORR and CRR were 52% and 24% respectively, compared to 37% and 16% for all treated patients [69]. Therefore, results are still pending for treatment-naïve, cisplatin- or platinum-ineligible mUC.

## 8. Discussion

### 8.1. What’s Next for Cisplatin and Platinum Ineligibility Criteria?

The current armamentarium for cisplatin- and platinum-ineligible patients is expanding with new developments in ADCs and targeted therapies. Yet, the need to determine patients’ eligibility to receive platinum compounds stems from the era of chemotherapy. Despite moving toward standardized definitions for both cisplatin and platinum ineligibility, the utility of this classification in clinical practice has become less pronounced in the new “post-EVP era”. This is mainly due to the early readout of EVP offering considerable and durable responses in both cisplatin-eligible and -ineligible populations, as shown in EV-302 and EV-103 Cohorts A and K, respectively [2,27,29]. Hence, this benchmark highlights the need for a newer clinical or molecular classification able to withstand the wave of non-chemotherapy drugs, from ADCs to FGFR inhibitors and innovative targeted therapies. 

Figure 2 visualizes this new change in the philosophy of treatment of LA/mUC in the “breaking the iceberg” model of mUC treatment. The “breakthrough fissure” at the tip of the iceberg represents the potential new SOC of EVP, while the two split poles represent the current dichotomy between cisplatin eligibility on one side and cisplatin/platinum ineligibility on the other. Classification based on cisplatin fitness is expected to become a limited approach in the current context of EVP. Beneath the surface lurks the value of molecular underpinnings as potential predictive biomarkers to classify patients with mUC. These variables could reinforce the current classification or replace it. They are related to either the tumor’s response to EVP or the mechanism of action of downstream therapies.

Beyond FGFR3 inhibitors as biomarker-informed therapies [12], and in the absence of other molecularly informed decisions, the subsequent-line setting will likely utilize chemotherapy in patients who remain fit and without significant neuropathy from EVP. Hence, classification by cisplatin or platinum ineligibility will likely be relegated to the subsequent setting. This expectation is also driven by the observed toxicity profile of EVP. Based on the reported AEs of EVP in the latest EV-302 update, physicians should be aware of the most frequent ones, including skin reactions of any grade (66.8%) and Grade ≥ 3 (15.5%), peripheral sensory or motor neuropathy (63.2%), ocular disorders (21.4%) and hyperglycemia (13%) [2]. While toxicity management is beyond the scope of this review, we review some treatment ‘pearls’ for these side effects. Early dose reduction of EV to 1 mg/kg or 0.75 mg/kg is warranted for neuropathy. Skin rashes can be handled by dose reduction and prescription of topical corticosteroid lotions, rather than ointments. Monitoring glucose levels at every dose administration and maintaining strict diabetes management are essential to prevent diabetic ketoacidosis. Although less frequent, cytopenias should be addressed by growth factor support. 

Adverse events in the phase 1/2b EV-103 trial were more common in the combination arm than in the monotherapy EV arm [26]. In EV-103 Cohort K, peripheral neuropathy was among the most common any-grade TRAEs in both the EVP and EV arms (60.5% vs. 54.8%, respectively). The same toxicity profile was also demonstrated by long-term follow-up in EV-103 Cohort A, with more than half of patients having any-grade peripheral neuropathy (any grade: 62.2%; grade ≥ 3: 4.4%) [29]. Although peripheral neuropathy might improve after holding off EV, it may also limit the ability to receive subsequent cisplatin- or carboplatin-based therapy. In this context, we expect a re-emergence of the importance of classification by cisplatin or platinum fitness, especially after long-term treatment with EV or upon progression on EV. The ability to control tumors that progress on EVP with cisplatin-based chemotherapy is yet to be determined. As kidney function is another important factor in this classification, EV has been shown to be less toxic to kidney function, with few reported cases of acute kidney injury (3%) and no adjustment needed by creatinine clearance [2,31].

The value of cisplatin or platinum eligibility may still be relevant in limited scenarios in the 1L setting, especially when EV is contraindicated or unavailable [89]. In EV-103 trials, the relatively high incidence of peripheral neuropathy and hyperglycemia in cohorts without uncontrolled DM or severe neuropathy suggest a possible limitation of EVP in patients with these risk factors. Additionally, the FDA prescribing information recommends avoiding the use of EV in patients with moderate (Child-Pugh Class B) or severe (Child-Pugh Class C) hepatic impairment [2,89].

### 8.2. The Value of Biomarkers

The new wave of mUC therapy requires a concurrent understanding of predictive biomarkers to tailor the right drug or combination to the right patient. In Table 4, we highlight the frequency of expression or alteration of select important biomarkers related to the drugs previously mentioned in the clinical trials. Some of these biomarker-guided goals were already set prior to EV, while others are now emerging after the establishment of EVP as the new SOC (Figure 3). This list of priorities is not exhaustive, as we will limit our discussion herein to the most prominent goals in both eras.

One goal from the pre-EVP era that is likely to continue into the post-EVP era is to understand the relevance of sequential and concurrent combinations of chemotherapy and immunotherapy in different treatment settings. It will be important to determine the role of immunotherapy in the post-EVP setting. The rationale behind this approach comes, at least in part, from the concept that chemotherapy causes tumor cells to release immunogenic tumor antigens. On one hand, it has been inferred based on the phase 3 Imvigor130, KEYNOTE-361, and DANUBE trials that platinum-based chemotherapy offers higher a ORR than ICIs, while CPIs might offer more durability, typically on the “tail end of the curve” [101]. Thus, initial chemotherapy for proper disease control followed by immunotherapy for long-term durability is a rational strategy. On the other hand, the positive results of combination nivolumab plus GC in CheckMate-901 hint at an additive effect between nivolumab and cisplatin, which is thought to be more immunogenic than carboplatin [48,90].

The use of PD-L1 as a predictive biomarker in all treatment settings should be abandoned. This biomarker was not consistent in predicting potential responders to ICIs in several trials with different PD-L1 assays and antibody clones [101]. Intertrial comparability is challenging due to a general inconsistency within the same test and the use of different tests to identify PD-L1+ tumors [101]. For example, the SP263 test identified 55% of patients as PD-L1+ in DANUBE, while SP142 identified only 23% in Imvigor trials. Staining intensity (SP263 and SP142 vs. DAKO 22C3), scoring algorithms, and cutoffs of PD-L1 have also been shown to be very heterogeneous compared to other markers [101].

Molecular profiling has also led to new targeted therapies that yield promising outcomes when combined with immunotherapy. FGFR inhibitors offer new options for patients with specific FGFR alterations (FGFR3 mutations and/or fusions). In the NORSE trial, common AEs of erdafitinib monotherapy included hyperphosphatemia (83.7%), stomatitis (69.8%) and diarrhea (41.9%) [39]. More research is needed to determine their role in sensitizing tumors to concomitant immunotherapy. Another obstacle in studying FGFR inhibitors lies in determining the optimal selection of patients with specific genetic alterations (mutations or fusions) to receive specific inhibitors of different isoforms [12]. For example, all patients in the NORSE trial had only FGFR3 alterations without FGFR2 alterations, but patients in the now-halted FIDES-02 trial had FGFR1–3 alterations [43]. Hence, optimizing assays to detect not only specific mutations but different alterations in isoforms of interest could give a better interpretation of the efficacy of the FGFR inhibitor in question. That would also entail the need for standardization of clinical testing for FGFR alterations in clinical practice, such as the use of next-generation sequencing (NGS) techniques in liquid biopsies [12].

Microsatellite instability, DNA mismatch repair status, and tumor mutational burden have also been associated with response to immunotherapy [13,102]. A classic example of the identification of relevant genetic signatures is the relation between high interferon gamma expression and response to nivolumab in CheckMate 275 [103]. Other identified biomarkers in the pretreated tumor microenvironment in this trial include CXCL9, CXCL10, CD8, and 12-chemokine signatures and tertiary lymphoid structures [103,104]. A meta-analysis of more than 1000 ICI-treated cases with exome and transcriptome data identified clonal TMB, total TMB, and CXCL9/CXCL13 expression as the strongest predictors of response to immunotherapy [105]. On the other hand, 9q34 (TRAF2) loss and CCND1 amplification predicted resistance to ICIs. Methylthioadenosine phosphorylase (MTAP) deficiency, due to chromosome locus 9p21 loss, is a trending biomarker with predictive and prognostic value [106]. In fact, MTA accumulation and metabolic vulnerability to de novo purine synthesis inhibition are two resultant aspects of MTAP deficiency that can be addressed by therapeutic applications [96,107]. APOBEC3, involved in cancer mutagenesis and clonal heterogeneity, is another promising, exploitable target to predict tumor response to chemotherapy and immunotherapy and restrict tumor progression [108,109].

While we review clinical biomarkers of relevance for the drugs mentioned earlier in the clinical trials, we highlight select molecular vulnerabilities with in vitro studies showing promising clinical applications. Whole-genome expression profiling of paired primary tumors and metastatic lymph nodes from surgical resection identified *FOXF1* as a differential expressed gene in the nodes, with a prognostic value (worse OS) [110]. In murine models of orthotopic xenografts, using human bladder cancer cell lines, *FOXF1* was found to be under-expressed in metastatic implants compared to primary cancers [110]. 

Genetic alterations in chromatin remodeling genes, such as *KMT2D*, *KDM6A* and *ARID1A*, represent frequent early events in urothelial neoplasia [111]. An example of a targetable frequent molecular aberration is the disruption of KDM6A, a key histone lysine demethylase, causing an epigenetic ‘switch’ to disrupt urothelial differentiation and promote neoplasia [112]. *KDM6A*-deficient bladder cancer cell lines show a loss of interaction with *FOXA1*, normally involved in urothelial differentiation, and a redistribution of transcription factor ATF3, further repressing *FOXA1*-target genes and activating cell cycle progression [112]. In mechanistic studies, KDM6A loss mediates EZH2-driven cellular proliferation and suggests a therapeutic application of inhibiting EZH2 methyltransferase to revese the effect [113]. Another epigenetic regulator frequently dysregulated in UC (18–25%), ARID1A, has not only shown altered signal transduction in other cancers, but also altered cell cycle control, DNA damage repair responses, tumor microenvironment and checkpoint signaling [111]. Based on the premise of *ARID1A* mutations increasing with disease staging, a recent work in cell lines with wild-type *ARID1A* (T24, SW1710 and 5637) has identified a more invasive phenotype with *ARID1A* knockout in the three lines [114]. Exploring the downstream molecular effects of these truncal epigenetic mutations, particularly in the treatment-naïve metastatic setting, could help identify novel predictive biomarkers. 

The post-EVP era raises the need for biomarkers related to the biology of complete or partial response to EVP [115,116]. Patients with initial progression on EVP, comprising around 7% to 9% of patients in EV-103 Cohort K and EV-302 [2,27], need to be analyzed for biomarkers or mechanisms of resistance through translational research using biospecimens collected from clinical trials. Early preclinical work in RT112 bladder cancer cell lines suggests that resistance to EV is mainly mediated by resistance to the MMAE payload and seems to be independent of the expression of surface target Nectin-4 [117]. Other biomarkers to predict or monitor the prevalent side effects of EVP, such as neuropathy and rash, are also of huge interest [118]. First, the ability to better predict and monitor side effects would help improve patients’ quality of life and maximize clinical efficacy during EVP treatment. Second, treatment de-escalation might be offered to spare patients long-term accumulating adverse effects such as neuropathy and enable them to receive subsequent cisplatin- or carboplatin-based regimens, known to be neurotoxic but with ancillary clinical utility and survival benefit in mUC [89]. 

Patient selection based on clinical and molecular characteristics will be another priority research directive in the post-EVP era. On the one hand, for responders, it is still unclear whether a combination strategy (administering EV and P together) or a sequential one is a better approach to consolidate the described deep and durable responses. On the other hand, some patients might not be eligible to receive the 1L combination, regardless of cisplatin or platinum eligibility. For example, the exclusion criteria of EV-103 included uncontrolled DM and ongoing sensory or motor neuropathy of grade ≥2 [119]. This fact might limit the extension of EVP to these patients. From a molecular standpoint, recent translational work quantifying Nectin-4 in matched primary and metastatic urothelial cancer samples identified not only a dynamic pattern of Nectin-4 expression, but also its decrease in a subset of metastatic biopsies associated with lower PFS [120]. However, the FDA review of EV-201 Cohort 1 noted the clinical efficacy of EV in all quartiles of Nectin-4 expression, despite a trend toward higher standardized expression scores in responders [121]. Therefore, there is a growing support for combining Nectin-4 expression with other composite biomarkers to predict response to EV or EVP and guide 1L treatment choices [122]. 

Another insightful example of patient selection in the metastatic space is shown in the prognostic relevance of specific metastatic sites in metastatic upper tract urothelial carcinoma (mUTUC). In a large cohort analyzed using the SEER database, patients with liver metastasis and multiple sites of visceral metastasis had poorer OS and cancer-specific survival that those with distant nodal disease [123]. Radical nephrouretectomy was also found to be associated with better survival, reinforcing the role of consolidative surgery after cisplatin-based chemotherapy [123]. As chemotherapy use and primary site-specific surgery predicted better survival in this population with mUTUC, aggressive treatment strategies could be justified in select oligometastatic patients.

Finally, dynamic molecular profiling before and after treatment can provide insight into mechanisms of resistance or new mutational signatures. One example of this challenge is to determine whether rechallenging patients with combinations such as EVP would be effective after failure of adjuvant immunotherapy. Based on phase 3 CheckMate-274, nivolumab was FDA approved in the adjuvant setting for high-risk patients after surgical resection, defined as having muscle-invasive disease (ypT2-T4a) and/or node-positive (N+) disease for patients with previous NAC, or extravesicular extension (pT3-T4a) and/or N+ disease for chemotherapy-naïve patients [124]. Biomarkers derived from the tumor microenvironment or time to failure of adjuvant immunotherapy may factor into such decisions.

In this context, personalized, tumor-informed approaches for molecular residual disease (MRD) detection are a powerful tool to guide escalation or de-escalation strategies. So far, the role of circulating tumor DNA (ctDNA) in predicting recurrence of bladder cancer has been well established in three settings. In the neoadjuvant setting, failure to clear ctDNA predicted recurrence better than pathological response at resection [125]. MRD evaluation has also been helpful in risk-stratifying patients who might be proper candidates for adjuvant therapy after cystectomy [125]. Third, while ctDNA assays have been proven to predict response to ICIs in the metastatic setting [126], it is yet to be seen whether the same approach could be replicated with EVP to de-escalate treatment. This is of particular importance in the newly identified subset of complete responders to EVP, to spare patients potentially additive adverse events. Moreover, all responders to EVP can benefit from the use of treatment-guiding ctDNA, which could be an earlier and more sensitive surrogate than radiographic response [125,126,127]. Hence, there is a predominant need to define molecular progression in the metastatic setting and compare it to our current SOC surveillance tools.

### 8.3. Practical Implications for Healthcare Professionals

While the outcomes of many investigated drugs in the 1L setting might not outperform the unprecedented survival benefit seen with EVP, some could show promise in subsequent treatment lines. Thus, next-generation sequencing (NGS) should be considered in all newly diagnosed patients, as identifying targetable alterations can guide 1L or subsequent lines of treatment. The position of these drugs in the treatment hierarchy depends on both the robustness of their predictive biomarkers, and the balance between their toxicity and the patient’s health status or quality of life.

EVP should be discussed as a 1L option for all cisplatin-ineligible and platinum-ineligible patients with LA/mUC. However, not all may receive this combination. First, some are not eligible for EV, due to uncontrolled DM (HbA1c ≥ 8%) and baseline ≥ Grade 2 peripheral neuropathy from previous systemic treatment. Others might be unfit for immunotherapy, due to autoimmune disorders, concomitant immunosuppressive agents, etc. In disadvantaged communities, many would still not be able to access or afford the drug combination. In these challenging cases, cisplatin-ineligible patients can be offered GCa and platinum-ineligible patients can be offered pembrolizumab monotherapy. 

For those declining these options, personalized treatment decisions should be formulated based on comorbidities, quality of life and patient preference. Genetic alterations identified through NGS testing can guide referral to clinical trials with relevant biomarkers. For example, bicycle toxin conjugates (BTCs), offering short duration of systemic exposure to the payload and a renal clearance, are interesting options for patients with hepatic dysfunction or baseline neuropathy. Patients with alterations in epigenetic modifiers, such as KDM6A and ARID1A, can benefit from trials of EZH2 inhibitors. Patients with tumors showing high microsatellite instability, high TMB or deficient mismatch repair, might benefit more from enrollment in immunotherapy trials [102]. Although not compared head-to-head with EVP in patients with FGFR alterations, the combination of erdafitinib plus cetrelimab in the NORSE trial showed an ORR of 54.5% and a median OS of 20.8 months [39]. 

During treatment with EVP, healthcare professionals should monitor and recognize early the development of common side effects, especially skin reactions, sensorimotor neuropathy, ocular disorders and hyperglycemic manifestations. For good responders who achieve disease stability, the feasibility of a maintenance strategy of immunotherapy, similar to the switch avelumab model after disease stability on GC or GCa in the ‘pre-EVP’ era, remains unknown. Exploring the addition of the prognostic and predictive values of ctDNA to radiographic staging is a reasonable direction in monitoring disease response.

After clinical progression or toxicity from EVP, exploring pan-FGFR mutation can give insights into subsequent eligibility for well-studied FGFR inhibitors such as erdafitnib. If no FGFR alterations are absent, healthcare professionals are advised to make an assessment of cisplatin or platinum-eligibility, now relegated to the subsequent setting. As in the ‘pre-EVP’ treatment indications, cisplatin-eligible patients should be offered ancillary cisplatin-based regimens. On the other hand, cisplatin-ineligible patients might benefit from GCa, while platinum-ineligible patients should be offered other antibody-drug conjugates such as sacituzumab govitecan. Choosing investigational drugs in subsequent lines can be guided by identified genetic alterations serving as predictive biomarkers. 

## 9. Conclusions

EVP will revolutionize our approach to treating LA/mUC, especially with its demonstrated deep and durable responses. Previously, cisplatin fitness was a major filter to decide 1L treatment. However, the superior responses of EVP will likely reduce the importance of this filter in the 1L setting and relegate it to the subsequent setting. Cisplatin or platinum ineligibility will likely be replaced or supplemented by other classifications based on eligibility for EV +/− immunotherapy or emerging molecular biomarkers. Research goals in the post-EVP era include identifying the optimal duration of EV treatment to avoid compromise between efficacy or durability and long-term side effects, such as peripheral neuropathy. Having fewer patients with neuropathy will minimize the proportion of cisplatin-ineligible patients at stable disease or upon progression on EVP. Physicians and healthcare professionals should monitor and recognize early the development of EVP side effects, especially skin reactions, sensorimotor neuropathy, ocular disorders and hyperglycemic manifestations. This monitoring should be accompanied by a frequent re-evaluation of the impact of treatment on the patient’s quality of life. For patients ineligible for or declining 1L EVP, physicians should advise ancillary chemotherapy regimens or personalized treatment strategies, based on drug toxicity and tumoral genetic alterations potentially serving as predictive biomakrers. Other goals in the post-EVP era include defining the mechanisms of resistance to EV and ICIs and refining patient selection for different treatment strategies based on molecular data. Many new and innovative drug classes are being studied across mUC subpopulations. Using patient-derived samples in these trials offers valuable information in translational research to understand the molecular underpinnings behind their mechanisms of action. 

## Figures and Tables

**Figure 1 biomedicines-12-00519-f001:**
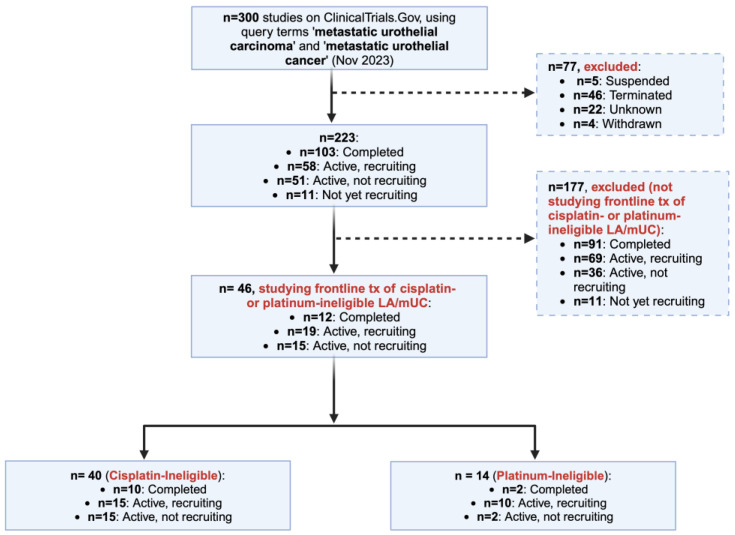
Flow diagram highlighting the stepwise approach of inclusion and exclusion of trials registered on ClinicalTrials.gov. Abbreviations: LA/mUC: Locally advanced or metastatic urothelial cancer; tx: treatment.

**Figure 2 biomedicines-12-00519-f002:**
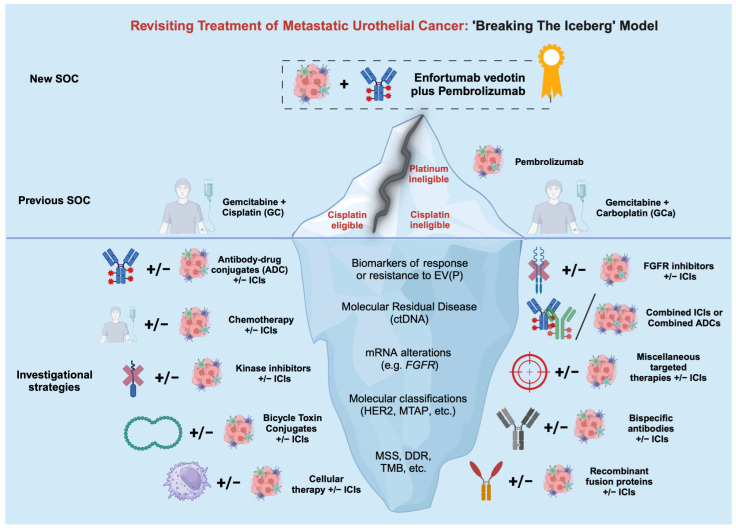
Our ‘iceberg model’ of the new treatment philosophy for mUC. Abbreviations: ctDNA: circulating tumor DNA; DDR: DNA damage response and repair; EV: enfortumab vedotin; FGFR: fibroblast growth factor receptor; HER2: human epidermal growth factor 2; ICIs: immune checkpoint inhibitors; MSS: microsatellite status; MTAP: methylthioadenosine phosphorylase; P: pembrolizumab; TMB: tumor mutational burden.

**Figure 3 biomedicines-12-00519-f003:**
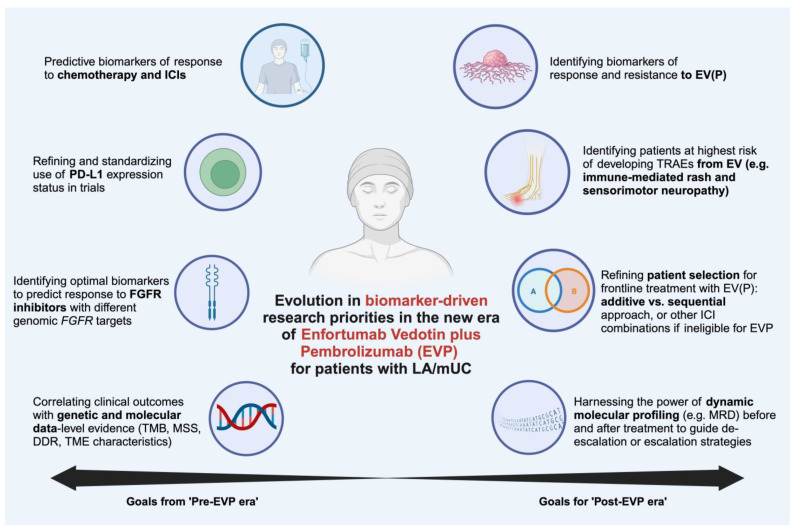
Evolution of research directions in biomarker analysis with the new SOC of EVP in mUC. Abbreviations: DDR: DNA damage response and repair; EVP: enfortumab vedotin + pembrolizumab; EV (P): enfortumab vedotin with or without pembrolizumab; FGFR: fibroblast growth factor receptor; ICIs: immune checkpoint inhibitor; MRD: molecular residual disease; MSS: microsatellite status; PD-L1: programmed death ligand 1; TMB: tumor mutational burden; TME: tumor microenvironment.

**Table 1 biomedicines-12-00519-t001:** Comparison of the consensus criteria for ineligibility for cisplatin- and platinum-containing regimens in mUC. Abbreviations: ECOG PS: European Cooperative Oncology Group performance status; KPS: Karnofsky Performance Scale; CrCl: creatinine clearance; NYHA: New York Heart Association; dB: decibels.

Parameters	Cisplatin Ineligibility (Galsky et al., 2011) [7]	Platinum Ineligibility (Gupta et al., 2022) [11]
ECOG PS	≥2, or KPS of ≤60%–70%	≥3
CrCl	<60 mL/min	<30 mL/min
NYHA Heart Failure Class	≥3	>3
Peripheral neuropathy	Grade ≥ 2 (i.e., sensory alteration or paresthesia, including tingling, but not interfering with activities of daily living)	Grade ≥ 2
Different parameters	Hearing loss (measured at audiometry) of 25 dB at 2 contiguous frequencies	ECOG PS of 2 and CrCl < 30 mL/min

**Table 2 biomedicines-12-00519-t002:** Summary of the most relevant clinical trials registered on clinicaltrials.gov by November 2023, investigating the combination of FGFR inhibitors and immunotherapy drugs in 1L treatment of cisplatin-ineligible LA/mUC. Abbreviations: AEs: Adverse Events; DLT: Dose-limiting toxicities; FGFRi: Fibroblast Growth Factor Receptor Inhibitor; ICI: Immune Checkpoint Inhibitor; NCT: National Clinical Trial; ORR: Overall Response Rate; OS: Overall Survival; PFS: Progression-Free Survival; TEAEs: Treatment-Emergent Adverse Events; TESAEs: Treatment-Emergent Serious Adverse Events.

NCT Number	Study Name	Most Current Study Status ^†^	FGFRi + ICI Combination?	Target of FGFR Inhibitor	Phase	Primary Outcome Measures
NCT03473743	NORSE	Active, not recruiting	Erdafitinib + cetrelimab	FGFR1-4	Ib/II	DLTs (Phase I), ORR and AEs (Phase II)
NCT04045613	FIDES-02 (Cohort 3)	Completed	Derazantinib + atezolizumab	Pan-FGFR	Ib/II	ORR, Safety, and tolerability of derazantinib alone and with atezolizumab
NCT03473756	FORT-2	Active, not recruiting	Rogaratinib + atezolizumab	FGFR1-4	Ib/II	DLTs, Number of subjects with TEAEs, drug related TEAEs, and TESAEs
NCT04003610	FIGHT-205	Terminated (business decision)	Pemigatinib + pembrolizumab	FGFR1-3	II	PFS

^†^ as updated on 3 November 2023 on ClinicalTrials.gov.

**Table 4 biomedicines-12-00519-t004:** Frequency of expression or alteration of select important biomarkers in the metastatic setting of urthelial cancer. Abbreviations: APOBEC: Apolipoprotein B mRNA Editing Catalytic Poly-peptide-like; Ephb2: Ephrin B2; Ephb4: Ephrin B4; FGFR: Fibroblast Growth Factor Receptor; HER2: Human Epidermal Growth Factor Receptor 2; HRR: Homologous recombination repair; MIBC: Muscle-invasive bladder cancer; MMR: Mismatch Repair; MSI-H: Microsatellite instability -High; MSS: Microsatellite stability; MTAP: Methylhyoadenosine phosphorylase; NMIBC: Non-muscle invasive bladder cancer; TCGA: The Cancer Genome Atlas; TLS: Tertiary Lymphoid Structures; TMB: Tumor Mutational Burden; UC: Urothelial Cancer; dMMR: deficient mismatch repair.

Biomarkers	Expression in UC as Clinical Interest	Alteration in UC as Clinical Interest	Details about Frequency of Expression or Alteration
Nectin-4	X		Frequently expressed in 83% [90]
Trop-2	X		Frequently expressed in ≤83% [91]
FGFR	X	X	FGFR3 most frequently expressed FGF receptor in normal urothelium [92]; *FGFR1-4* alterations in 33% [12]
Ephb-2/ Ephb4	X		Extremely low Ephb2 expression (~nil) and high Ephb4 expression in 94% [93]
HRR genes (*BRCA1*, *BRCA2*, *ATM*, *CDK12*, etc.)		X	HRR mutations identified in 31.4% of a TCGA cohort (*n* = 822) and 34.1% of a retrospective single-center cohort (*n* = 343) [94]
HER2	X	X	Expression in 6–37% Alteration in 12% [95]
MTAP		X	Loss in 28% [96]
APOBEC		X	Mutation signature in 80% [97]
MSS/MMR		X	Prevalence of dMMR in 6% of UC and 2% in BC; Prevalence of MSI-H in 3% of UC and 1% in BC [98]
TMB		X	High TMB in 26% of Stage II-Stage IV BC, low TMB in 74% of cases [99]
TLS	X		~25% of NMIBC and ~75% of MIBC [100]

## Data Availability

The datasets generated and analyzed in the current study are available in the ClinicalTrials.gov Protocol Registration and Results System (PRS) repository. Web link: www.clinicaltrials.gov (Last Accessed on 3 November 2023).

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
