# Peer review of "Revisiting Treatment of Metastatic Urothelial Cancer: Where Do Cisplatin and Platinum Ineligibility Criteria Stand?"

_biomedicines, 2024, doi:10.3390/biomedicines12030519_

Round 1
Reviewer 1 Report
Comments and Suggestions for Authors
1. The article's title might benefit from increased specificity. A clearer representation could be achieved with a title like "Reassessing Advanced Urothelial Cancer Treatment: Approaches for Platinum-Incompatible Drugs."
2. Enhancing the article's structure for better clarity is advisable. Employing subheadings or paragraph markers can effectively delineate various sections, facilitating a more comprehensive understanding of the content.
3. Within the discussion section, a more in-depth analysis of the practical application of diverse treatment strategies in clinical settings and guidance on selecting optimal treatment plans based on individual patient scenarios would be valuable.
4. Expanding the article's discussion on the role of predictive biomarkers and their application in steering treatment strategies could offer a more comprehensive understanding of these critical components.
5. Further exploration of methods to refine patient screening processes for leveraging emerging molecular biomarkers and treatment strategies could enhance the article's depth and practical relevance.
6. Providing additional details on drug side effects and safety considerations would enable readers to grasp the potential risks associated with various treatment options, contributing to a more informed perspective.
7. To conclude, offering additional recommendations on the practical implementation of these novel treatment strategies would assist healthcare professionals in delivering more personalized and effective care to patients.
Comments on the Quality of English Languagesee the comments
Author Response
We thank the esteemed reviewer for their valuable feedback. Please find below a point-by-point reply to the mentioned suggestions. Kindly refer to the attachment to find mentioned text changes highlighted in grey:
- We dearly appreciate our reviewer's suggestion, yet we would like to explain our rationale for maintaining the current title. The terms 'cisplatin ineligibility' and 'platinum ineligibility' are standard terminologies in genitourinary oncology, especially in the treatment of metastatic urothelial cancer. Our review aims to update the GU oncologist on the latest updates in these two patient subpopulations with previous major unmet needs. Thus, the term 'platinum-incompatible drugs' may not fully capture our focus of defining this new treatment philosophy. We kindly suggest referring to the following work by other colleagues and leaders in the field for context: https://doi.org/10.1038/s41571-023-00826-2; https://doi.org/10.1016/j.medj.2023.11.010
- The article's updated structure has been revised with less subheadings for a smoother read and better clarity. However, we apologize for not being able to employ paragraph markers to abide by the standardized template format of the journal.
- We thank the reviewer for bringing our attention to this insightful addition. To make this topic more easily identifiable by the interested GU oncologist or healthcare professional, we dedicate a separate section in our Discussion (8.3) to integrate predictive biomarkers, clinical applications and clinical reasoning in the sphere of 1L treatment of cisplatin- and platinum-ineligible mUC.
- Kindly refer to Point 3.
- Kindly refer to Point 3.
- A detailed description of the most common side effects of the drugs of interest was added to our manuscript.
- Practical details and recommendations for healthcare professionals have been now added to the Conclusion section of the manuscript.
We'd also like to bring the attention of our reviewer to the following changes:
- Some text and captions for newly added tables and figures are distinguished by yellow and blue highlights. These revisions incorporate feedback from other peer reviewers, aiming to enhance the clarity and clinical utility of our manuscript. We hope these changes contribute further to the overall quality of the content.
- Our reviewer suggested required moderate editing go the English language of our manuscript. To meet the high standards of the journal and satisfy the judgment of our reviewer, the manuscript has been addressed for language editing by Ms. Sarah Townsend, Senior Scientific Editor at our institution. We thank her efforts in contributing to this refined version in the Acknowledgment section.
We thank the reviewer again for their time and attention and we hope our changes meet their expectations.

Reviewer 2 Report
Comments and Suggestions for Authors
Manuscript entitled "Revisiting Treatment of Metastatic Urothelial Cancer: Where do Cisplatin and Platinum-Ineligibility Criteria Stand?"
This work is of interest while certain modification should be made before it can be accepted:
1. The authors should summary the related biomarkers and the freqency of alteration in a table.
2. The authors should mention selected in vitro evidence for more support.
Comments on the Quality of English LanguageAcceptable.
Author Response
We thank our esteemed reviewer for their kind appreciation of the work and valuable feedback. Please find below a point-by-point reply to the mentioned suggestions:
- We highly appreciate the value of this added table to our manuscript. Please find a new Table 4 summarizing the frequencies of alteration or expression of biomarkers related to our mentioned drugs. The changes are highlighted in yellow in the attachment for the convenience of our reviewer.
- As the authors agree with this valuable input, the authors have now added a dedicated paragraph explaining selected in vitro evidence, highlighted in yellow as well.
We'd like to bring the attention of our reviewer to the following changes:
- Text highlighted in blue reflect other changes made upon suggestions of other reviewers.
- Our reviewer suggested a minor edit to the English language of our manuscript. To meet the high standards of the journal and satisfy the judgment of our reviewer, the manuscript has been addressed for language editing by Ms. Sarah Townsend, Senior Scientific Editor at our institution. We thank her efforts in contributing to this refined version in the Acknowledgment section.
We thank our Reviewer again for their time and attention and we hope our changes meet their expectations.

Reviewer 3 Report
Comments and Suggestions for Authors
I read with great interest this review on Revisiting Treatment of Metastatic Urothelial Cancer.
minor suggestions:
1) The material and methods section should be improved. Define the type of review. Add a figure on the included/excluded studies (e.g. PRISMA diagram if systematic review).
2) Patient selection is a priority research directive in the post-EVP. Consider adding a few lines in the discussion on the role of organ-specific metastases in UTUC. Authors may rely on https://doi.org/10.3390/jcm11185310.
Author Response
We thank our esteemed Reviewer for their kind appreciation and valuable feedback. Please find below a point-by-point reply to the mentioned suggestions:
- The Material and Methods section (highlighted in yellow) has been refined and re-drafted for clarity and transparency. Moreover, a Flow Diagram (now as 'Figure 1') has been also added to show the stepwise approach to include and exclude studies of interest for our narrative review.
- As we appreciate our Reviewer's suggestion to tackle this crucial point in patient selection, a new paragraph detailing this topic has been now added to the Discussion section and highlighted in yellow.
- Our reviewer rated the correctness and readability of the English language in our manuscript with 3 out of 5 stars. To meet the high standards of the journal and satisfy the judgment of our reviewer, the manuscript has been addressed for language editing by Ms. Sarah Townsend, Senior Scientific Editor at our institution. We thank her efforts in contributing to the refined version of the manuscript in the Acknowledgment section.
We thank our Reviewer again for their valuable time and attention, and we hope our edits meet their expectations.
